# Impact of perceived factors of coronavirus infection on COVID-19 vaccine uptake among healthcare workers in Ghana—Evidence from a cross-sectional analysis

Emmanuel K. Gelyi[1], John Azaare [ID][2]*, Nana Kobea Bonso [ID][3], Mary Rachael Kpordoxah[4], Gifty Apiung Aninanya[2]

1 Department of Social and Behavioural Sciences, School of Public Health, University for Development Studies, Tamale, Ghana, 2 Department of Health Services, Policy Planning, Management and Economics, School of Public Health, University for Development Studies, Tamale, Ghana, 3 Department of Engineering, School of Engineering, University for Development Studies, Tamale, Ghana, 4 Department of Global and International Health, School of Public Health, University for Development Studies, Tamale, Ghana

* jazaare@uds.edu.gh

## Abstract

### Introduction

Ghana faced acute COVID-19 vaccine uptake rejection after the rollout of the initial dose, thus, posing a risk of not reaching herd immunity as necessary to curb the spread of the novel coronavirus disease.

### Objective

In this study, we analysed the impact of perceptions of the COVID-19 infection on COVID-19 vaccine uptake among healthcare workers in the Mampong district of Ghana.

### Methods

The study was conducted between April 2022 and June 2023 and interviewed 260 respondents using a closed-ended electronic questionnaire in a Google form format. We then analysed for association using a composite outcome response of healthcare workers in Ghana using a multiple logistics regression model. The alpha value was set at p < 0.05 for statistical significance employing statistical software, IBM Statistical Package for the Social Sciences software. The analysis adjusted for independent covariates using respondent medical history, COVID-19 infection status, and sociodemographic characteristics.

### Results

Out of the total respondents, 219 (84.2%) took at least one shot of a COVID-19 vaccine. Of those who took a vaccine, 61.9% took AstraZeneca, followed by Johnson and Johnson (8.5%) and Pfizer BioNTech (6.2%). Vaccine uptake was significantly associated with positive previous vaccination history (p < 0.001), perceived vaccine safety (p < 0.001),

**Data availability statement:** Access to the data used in this analysis is restricted by the School of Public Health, University for Development Studies, Tamale, Ghana and not publicly available. However, minimal data set can be accessed upon request to the Graduate School of the University for Development Studies via graduateschool@uds.edu.gh.

**Funding:** The author(s) received no specific funding for this work.

**Competing interests:** The authors have declared that no competing interest exist.

perceived seriousness of COVID-19 infection (p < 0.008), and trust in COVID-19 vaccine based on recommendations by experts (p < 0.015).

## Conclusion

Previous vaccination history and perceived factors such as vaccine safety, the seriousness of the COVID-19 infection, perceived risk of infection, and trust in expert recommendations influenced vaccine uptake among healthcare workers in Ghana.

## Introduction

According to the World Health Organization (WHO), COVID-19 is no longer a pandemic and is no longer considered a disease of public health emergency. However, COVID-19 infection-related deaths are still recorded globally [1]. As of October 18, 2023, 0.77 billion infections have been confirmed, with nearly 7 million deaths worldwide [2].

Europe and the Western Pacific were the hardest hit during the pandemic peak periods, with confirmed cases in the region of 276 million and 207 million respectively. Although Africa recorded relatively low figures as compared to other parts of the world, the numbers nonetheless were staggering, with nearly 13 million positive cases and 258,562 deaths by the time the pandemic was no longer considered a disease of public health emergency [2]. Ghana confirmed its first case of COVID-19 in March 2020 [3] and reported 171,160 laboratory-confirmed cases as of February 20, 2023 [4].

Following the insurgence of the novel coronavirus, the scientific community managed to produce a vaccine that has now proven useful in curbing the spread of COVID-19 or at least mitigating the severity of the virus. Although COVID-19 is no longer considered a disease of public health emergency, vaccine distribution and uptake remain a public health interest [5]. As of October 23, 2023, 13.5 million vaccine doses have been administered worldwide [1]. The vaccine has proven helpful in slowing the infection transmission since the first dose was inoculated in New York [6]. Given the initial limited supply of vaccines in the global community, particularly in lower- and middle-income countries, a select few had the privilege of being vaccinated first.

Healthcare workers were among the initial beneficiaries of the COVID-19 vaccination project globally, considering their frontline role in combating COVID-19 particularly during the peak periods [7]. Given the peculiarity of their work, healthcare professionals suffered infection as well, and literature has shown that they were among some of the most affected groups since the emergence of the pandemic, thus, close attention was given to them to boost their chances of a win in the fight against the pandemic.

At the peak of the pandemic, nearly 3000 health workers were infected in Ghana, resulting in 11 deaths [8]. Literature suggest that Ghana had vaccinated 38.40% of its citizens as at January 15, 2023, yet, this was not without controversies and myths on COVID-19 vaccine uptake [9,10]. As a consequence, vaccine hesitancy persisted even among healthcare workers, thus posing a threat to vaccine uptake and the chances of achieving herd immunity [11].

Elsewhere in South Africa and Zimbabwe, studies have shown that less than 50% of the targeted population were willing to take up the COVID-19 vaccine, citing doubts about the vaccine's usefulness and safety [11,12]. Other studies reported mixed views of COVID-19 vaccine acceptance among health professionals in Ghana [13,14]. Whereas the literature on COVID-19 vaccine uptake remains scanty and narrowed, vaccine uptake despondency among trained healthcare workers is also worrying. In March 2022, an analysis from the Ghana District Health Information Management System (DHIMS) showed that out of the

31,238 members of the general population in the Mampong municipality of Ghana who took the first dose of the COVID-19 vaccine, 19,237 people including healthcare workers failed to take a second dose, despite their direct involvement in the vaccination programmes and campaigns.

At the emergence of the coronavirus pandemic, speculations and conspiracy theories, particularly in lower- and middle-income countries, varied constantly and coupled with the limited knowledge of COVID-19 at the its early stages, vaccine uptake probably is negatively impacted..

Therefore, this study aimed to assess perceived factors affecting COVID-19 vaccine uptake among healthcare workers in Ghana. In the current approach, we adapted an existing framework constructed by Azaare *et al* (2020) [15] to design the current study to determine the association between COVID-19 vaccine uptake and perceived factors related to COVID-19 infection such as the perceived seriousness of COVID-19, risk of infection, trust in the recommendation of experts, vaccine country of origin, perceived vaccine effectiveness and perceived vaccine safety, while accounting for confounding factors (Fig 1).

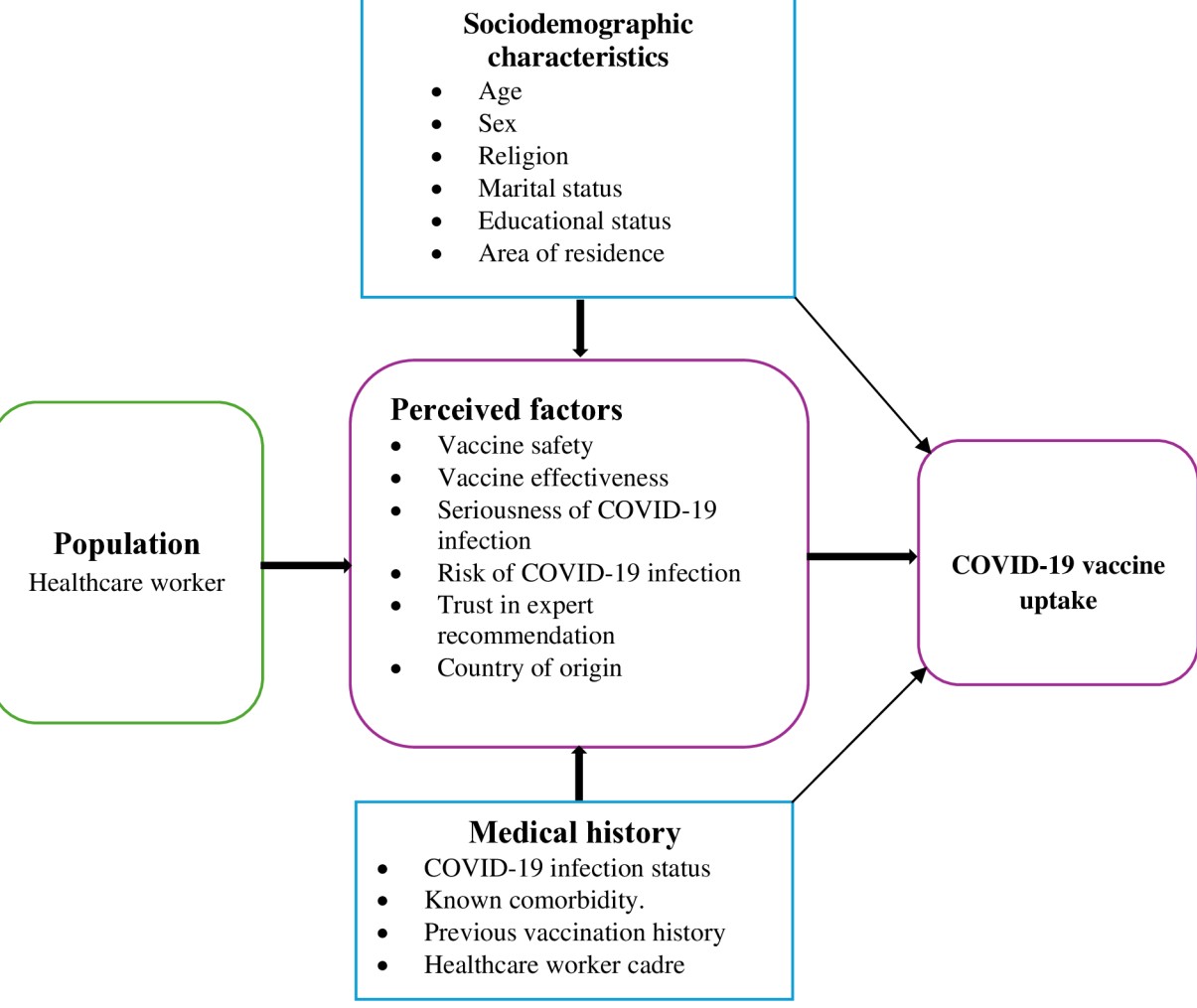

**Fig 1. Mitigation factors of COVID-19 vaccine uptake (adapted from Azaare *et al.,* 2020).**

## Materials and methods

### Study design

The study used a cross-sectional design and collected one-time data from healthcare workers in Mampong municipality of the Ashanti region of Ghana between April 24 and June 23, 2022. The study considered the uptake of the COVID-19 vaccine among healthcare workers across different cadres (Table 1) as a composite dependent variable. We then determined the factors of association listing common perceived factors such as perceived seriousness of COVID-19 infection, perceived vaccine safety, perceived country of origin, perceived risk of COVID-19 infection and trust in expert recommendation by the WHO or Ghana's Ministry of Health (MOH). Data on participants' previous medical history and sociodemographic characteristics, such as age, gender, religion, marital status, educational status, and area of residence were collected from the participants using an online Google form, and these were adjusted for in multiple logistic regression model (Fig 1).

### Study settings

Mampong municipality is a typical Ghanaian district which is predominantly Christian and relies on the Primary Health Care (PHC) concept and the Community-based Health Planning and Service (CHPS) initiative described as level 3 and level 4 categories of care respectively. The district is home to about 107,331 people according to the 2021 population and housing census. Mampong is bordered to the south by Sekyere-Dumasi, to the east by Sekyere South Municipal, and to the north by Sekyere Central District and has six PHC health centres and five CHPS compounds. The Mampong township also has five private health service providers with a health workforce of about 670.

### Sample size determination

Yamane's formula for random sampling depicted as n = $N/(1 + Ne^2)$ was used in determining the sample size, given the known population size. Thus, n = number of samples, e = margin of error (5%), N = population size (670 health care workers). Therefore, $650/1 + 650 (0.05^2)$ = 250. We then estimated a 5% non-response rate and approximated a sample size of 262.

### Inclusion and exclusion criteria

Healthcare workers 18 years and above and residents in the Mampong municipal were included in the study if they consented to participate. Healthcare workers who reported ill were excluded during the data collection process.

### Data collection

We developed a structured self-administering questionnaire using Google surveys form and collected data electronically from the study participants. The health facilities within the municipality were grouped into strata, and the number of participants in each stratum was determined using a proportionate stratified random sampling technique. A simple random sampling method was then used to obtain the participants from each stratum using an equal number of folded pieces of paper with 'yes' or 'no' presented to eligible participants. Those who chose 'yes' were selected for the study and if they consented, a link to the questionnaire was shared with them in email, WhatsApp, Facebook or via mobile phone or computer. Respondents who required technological assistance were given. The Questionnaire (attached as supplementary file 1) focused on respondents' perceptions of the risk of COVID-19 infection, perception of the seriousness of COVID-19, COVID-19 vaccination status, brand of COVID-19 vaccine, expert recommendation, medical history, and participants' sociodemographic characteristics.

**Table 1. Respondents COVID-19 infection status and socio-demographic characteristics.**

| Description | Frequency (n) |
| --- | --- |
| **Age** | |
| 20–29 | 96 (36.9%) |
| 30–39 | 136(52.3%) |
| 40–49 | 19(7.3%) |
| 50–59 | 9(3.5%) |
| **Gender** | |
| Male | 109(41.9%) |
| Female | 151(58.1%) |
| **Educational status** | |
| Primary | 3(1.2%) |
| Secondary | 24(9.2%) |
| Tertiary | 333(89.6%) |
| **Area of residence** | |
| Rural | 67(25.8%) |
| Urban | 193(74.2%) |
| **Healthcare worker cadre** | |
| Clinicians | 23(8.8%) |
| Biomedical scientist | 6(2.3%) |
| Nurse/midwife | 160(61.5%) |
| Dispensary technician | 12(4.6%) |
| Allied health professional | 18(6.9%) |
| Other professionals | 41(15.8%) |
| **Marital status** | |
| Single | 123(47.3%) |
| Married | 133(51.2%) |
| Co-habitation | 2(0.8%) |
| Widow/widower | 2(0.8%) |
| **Religion** | |
| Christianity | 249(95.8%) |
| Islam | 10(3.8%) |
| African Traditional Rel. | 1(0.4%) |
| **Known comorbidity** | |
| Yes | 23(8.8%) |
| No | 237(91.2%) |
| **COVID-19 infection** | |
| Yes | 25(9.6%) |
| No | 221(85.0%) |
| Not sure | 14(5.4%) |

(Source: Field Data, 2022).

## Statistical analysis

Responses were retrieved into Microsoft Excel version 19 and cleaned and subsequently transferred to SPSS version 25.0 for analysis. The analysis was examined for frequencies and proportions of the respondents' characteristics. We also analysed for association between COVID-19 vaccine uptake and respondents' socio-demographic characteristics such as age, gender, religion, marital status, education, area of residence, COVID-19 infection status,

known comorbidity, and healthcare worker cadre using chi-square test statistic. We then adjusted for confounding using multiple logistic regression and checked for statistical significance, p < 0.05. The Manuscript was written and reported using the SQUIRE 2.0 checklist guidelines [16].

### Ethics approval and consent to participate

This study first received approval from the graduate studies academic board of the School of Public Health, University for Development Studies, Tamale. The study further received ethical clearance from the Kwame Nkrumah University of Science and Technology Ethical Review Committee and referenced CHRPE/PA/142/22. Additionally, a formal letter of request for admittance was sent to the health directorate of the study site for permission to conduct the study. All respondents consented to participate in this study by responding to the Google questionnaire asking for their consent before gaining access to respond to the rest of the questionnaire. Respondents who did not consent were denied access to continue to respond to the questionnaire. No witnesses were required considering all respondents were above 18 years of age and considered adults as per the 1992 Constitution of the Republic of Ghana. Respondents retained their right to withdraw from the study at any time without prior notice to the researchers. The study design concealed respondents' identities, and their opinions and values maintained.

## Results

### Respondents' sociodemographic characteristics and COVID-19 vaccine uptake

In all, 260 out of 262 respondents returned their questionnaire showing a 98.8% response rate. Out of the total respondents (n = 260), 219 (84.2%) took at least one shot of a COVID-19 vaccine (Table 1). All respondents were in active service, i.e., below 60 years in the public sector in Ghana. Nine out of ten respondents were less than 40 years of age. Out of the total respondents, 151 (58.1%) were females, and 109 (41.9%) were males. A little over half of the respondents were married (51.2%), and nearly 90% had tertiary education. Respondents were mainly nursing care professionals, 169 (61.5%). Of all the respondents, 23 (8.8%) reported having a known chronic illness, while one out of ten said they have tested positive for COVID-19 infection since the pandemic outbreak in Ghana in March 2020.

### COVID-19 vaccination experience

Of those who took the COVID-19 vaccine, 61.9% took AstraZeneca (oxford brand) vaccine, followed by Johnson and Johnson (8.2%). Sputnik V was the least vaccine taken among the study participants (3.5%) (Fig 2). Most respondents reported side effects ranging from severe headache (54.8%), generalised malaise (45.8%), injection site swelling (23.1%) and fever (22.7%) (Fig 3). Out of those who experienced side effects, the majority considered theirs as mild, 103 (47.0%), followed by very severe 46(21.0%), moderate, 43 (19.6%), and then severe, 27 (12.3%) (Table 2).

### Perceptions of COVID-19 infection and vaccine safety among healthcare workers

Overall, 85.8% of respondents agreed that they are at risk of COVID-19 infection due to occupational exposure. Almost three-quarters (73%) believe that they can protect themselves from COVID-19 and do not require the vaccine. More than half (64.6%) agree that their

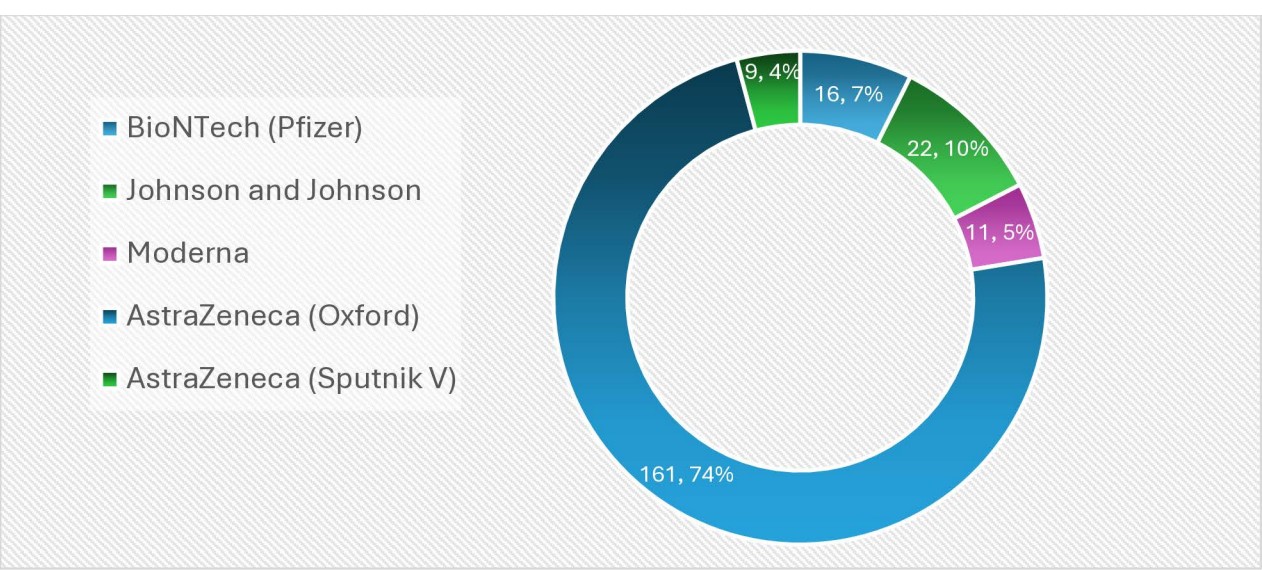

**Fig 2. COVID-19 vaccines received by healthcare workers by brand.**

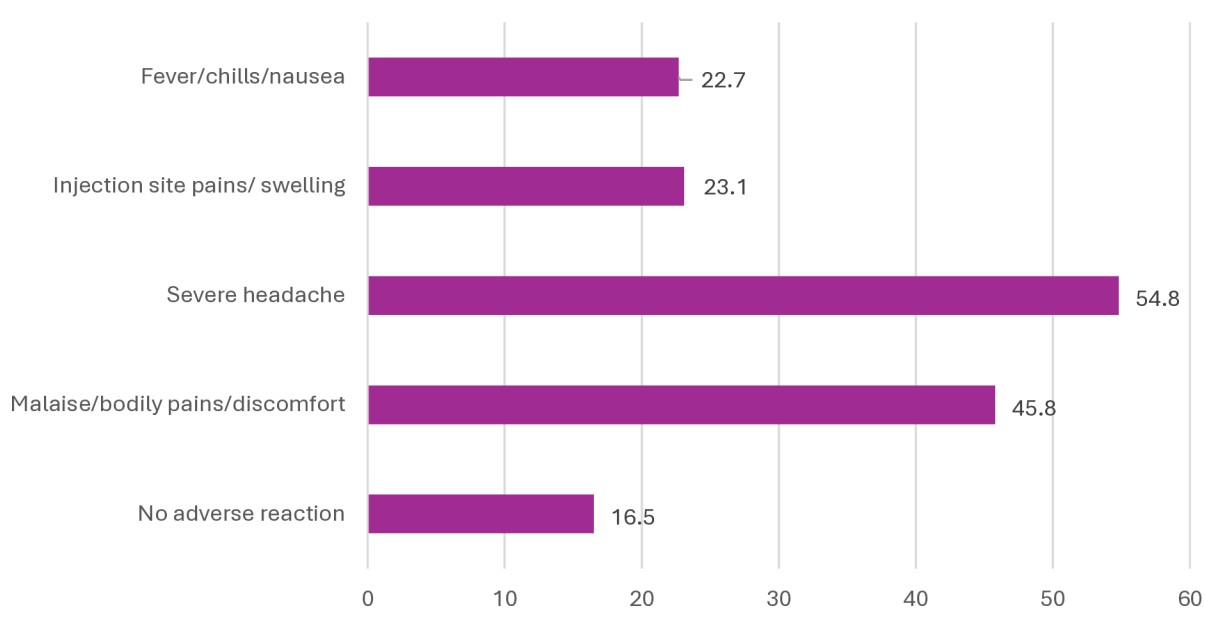

**Fig 3. Post COVID-19 vaccination side effect.**

families, patients, and friends will be protected if they took a vaccine. Two-thirds (60%) of the total respondents have confidence in the measures Ghana's Ministry of Health put in place to control the pandemic and have trust in the COVID-19 vaccine recommendation by the WHO and the Ministry of Health. On vaccine safety, 60% agree that the vaccine is safe, and 50.4% agree the vaccine is effective. Nine out of ten participants, (88.1%) do not believe that COVID-19 vaccine had a hidden agenda. However, more than 2/3 of respondents were concerned about the likely side effects of the COVID-19 vaccine. Nonetheless, most of the study participants (85.4%) would recommend the COVID-19 vaccine to eligible individuals. Most of the

**Table 2. Respondents COVID-19 vaccination status and reported side effects.**

| Variable | Frequency (n) |
|---|---|
| *COVID-19 Vaccination status* | |
| No | 41(15.8%) |
| Yes | 219(84.2) |
| Total | 260(100%) |
| *No. of dose taken* | |
| 1st dose | 83(37.8%) |
| 2nd doses | 114(52.0%) |
| Boster dose | 22(10.0%) |
| Total | 219(100%) |
| *Reported side effects* | |
| No | 42(16.1%) |
| Yes | 218(83.8%) |
| Total | 260(100%) |
| *Severity of side effect* | |
| Mild | 103 (47.0%) |
| Moderate | 43 (19.6%) |
| Severe | 27 (12.3%) |
| Very severe | 46(21.0%) |
| Total | 219(100%) |

study participants disagree with the notion that vaccines developed in Europe and America are safer than those developed in other countries and disagree that vaccines deployed in Africa and Ghana are less effective and less safe, with 43.5% and 48.8%, respectively (Table 3).

## Association between perceived factors and COVID-19 vaccine uptake among healthcare workers

Comorbidity and cadre of healthcare worker were found to be statistically significant in association with COVID-19 vaccine uptake among the healthcare workers; p = 0.001 and p = 0.030, respectively (*supplementary sheet 2*). Perception of COVID-19 vaccine: aOR = 0.048, 95% CI (0.715, 2.763); p = 0.006, previous vaccine uptake: aOR = 0.048, 95% CI (0.715, 2.763); p = 0.006, perceived vaccine safety: aOR = 0.126, 95% CI (0.027,0.373); p = 0.001, perceived seriousness of COVID-19 infection: aOR = 0.077, 95% CI (1.75,2.934); p = 0.008, and trust in experts' recommendation aOR: = 0.048, 95% CI (1.250,7.704); p = 0.015 were statistically significant in associated with COVID-19 vaccine uptake among the healthcare workers. However, the association between COVID-19 vaccine uptake and COVID-19 infection status, perceived vaccine effectiveness, vaccine country of origin, and perceived difference of Africa-allocated vaccines were statistically not significant (Table 4).

## Discussion

Nurses and midwives were the major respondents in this study, constituting nearly two-third of the study participants. Although the sampling may have skewed unfavourably across all cadres in the study area, it also presents some leverage to interpret the results towards essential healthcare cadres, given the significant role of nursing staff during the pandemic peak period. Respondents were nearly all Christians, 95.8%, and this spoke to the nagging perceptions of myths and beliefs about COVID-19 and vaccine uptake. Less than 10 percent tested positive

**Table 3. Health care workers perception of COVID-19 infection and vaccination.**

| Variable | Agree n (%) | Disagree n (%) | Not Sure n (%) |
|---|---|---|---|
| COVID-19 is a serious disease | 177 (68.1) | 57 (21.9) | 26 (10) |
| I'm at risk of COVID-19 due to occupational exposure | 223 (85.8) | 21 (8.1) | 16 (6.2) |
| I do not need vaccine to protect myself from COVID-19. | 40 (15.3) | 190 (73.1) | 30 (11.5) |
| Being vaccinated can protect my family, and friends. | 168 (64.6) | 33 (12.7) | 59 (22.7) |
| Vaccines produced in the USA and Europe and USA safer | 50 (19.2) | 113 (43.5) | 97 (37.3) |
| Vaccines are less effective and less safe. | 53 (20.4) | 127 (48.8) | 80 (30.8) |
| COVID-19 Vaccine is safe | 156 (60) | 26 (10) | 78 (30) |
| COVID-19 vaccine is effective | 131 (50.4) | 55 (21.2) | 74 (28.5) |
| COVID-19 vaccine side effect is a concern to me | 179 (68.8) | 48 (18.5) | 33 (12.7) |
| I have confidence in COVID-19 control measures | 156 (60) | 85 (32.7) | 19 (7.3) |
| I have previously been vaccinated as an adult. | 240 (92.3) | 20 (7.7) | |
| COVID-19 vaccines have a hidden agenda | 31 (11.9) | 229 (88.1) | |
| I have uncertainty about the COVID-19 vaccine safety | 136 (52.3) | 124 (47.7) | |
| I trust in the COVID-19 vaccine recommended by WHO | 156 (60) | 104 (40) | |
| I will recommend COVID-19 vaccine to eligible individual | 222 (85.4) | 10 (3.8) | 28 (10.8) |

(Source: Field Data, 2022).

**Table 4. Logistics regression analysis of association health care worker COVID-19 vaccine uptake and perceived factors and socio-demographic characteristics.**

| Variables | aOR | 95% (Conf. Interval) | P value |
|---|---|---|---|
| Age | 0.408 | (0.261–0.085) | 1.949 |
| Sex | 0.885 | (0.796–0.350) | 2.236 |
| Marital status | 2.595 | (0.056–0.974) | 6.915 |
| Residence | 0.611 | (0.362–0.211) | 1.764 |
| Religion | 0.155 | (0.089–0.018) | 1.325 |
| Educational level | 2.275 | (0.368–0.381) | 13.603 |
| Category of HCW | 1.458 | (0.571–0.396) | 5.360 |
| Perception of COVID-19 vaccine | 0.048 | (0.124–0.745) | **0.006*** |
| Previous vaccine uptake | 0.101 | (0.027–0.373) | **0.001*** |
| COVID-19 vaccine is safe | 0.126 | (0.715–2.763) | **0.001*** |
| At risk of COVID-19 | 0.243 | (0.070–0.844) | **0.026*** |
| COVID-19 is serious | 0.077 | (1.175–2.934) | **0.008*** |
| Trust in experts' recommendation | 0.048 | (1.250–7.704) | **0.015*** |
| Can protect self against COVID-19 | 0.137 | (0.194–1.253) | 0.493 |
| Perceived vaccine is Effectiveness | 0.945 | (0.246–3.633) | 0.934 |
| Vaccine is a way to prevent and control COVID-19 | 0.931 | (0.268–3.229) | 0.910 |
| Vaccine will protect family, patient, and friends | 0.543 | (0.178–1.654) | 0.282 |
| Vaccines produced in the USA and Europe are safer | 1.257 | (0.456–3.467) | 0.659 |
| COVID-19 Status | 1.580 | (0.347–7.194) | 0.554 |

*Statistically significant at p < 0.05.

for COVID-19, more than one year since the COVID-19 was reported in Ghana and when the first Pfizer-BionTech COVID-19 Vaccine was released.

The results show that vaccine uptake was high among the healthcare workers in the study area (84.2%), far more than the expected target of 70%. Oxford AstraZeneca brand was the leading brand taken by healthcare workers, and this is consistent with earlier evidence in Ghana, Greece, the United States and Hong Kong [17,18]. This was also not surprising as AstraZeneca was the first COVID-19 vaccine to be distributed by the COVID vaccine global access (COVAX) [19]. AstraZeneca was consciously distributed given its availability and recommendation from world bodies [20,21] and probably explained its popularity among the vaccinated brands. Given an option, healthcare workers probably would go in for other brands, other than AstraZeneca.

Our findings also show an improvement in vaccine uptake among the study population relative to earlier studies, which found 38.3% [22,23], showing a glimpse of hope of vaccine acceptability albeit on a gradual scale. It also means the general population can sooner rely on the examples of the health workers and take up the vaccine to boost the country's chances of herd immunity. Population COVID-19 vaccine uptake adherence is significant in determining the overall prospect of achieving herd immunity. Further, the vaccination uptake rate among healthcare workers is relevant in improving the odds of population vaccine acceptability as the healthcare workers show willingness to take up the vaccine themselves.

The current findings also show a substantial improvement in vaccine uptake over time, in contrast with a web-based cross-sectional study conducted in Nigeria, which showed that only 20% of the participants were willing to participate in the COVID-19 vaccination exercise [24]. The disagreements in the findings may be due to differences in the study period, particularly as at the early stages of vaccine development, the prospects of the COVID-19 vaccine were uncertain. It is also safe to argue that the willingness to take up a vaccine may differ from actual vaccine uptake. Moreover, most of the factors identified in previous studies might have been addressed or modified and perhaps influenced the outcome of the current results, hence an improvement over time relative to vaccine uptake. Nevertheless, it is important to note that even though this study did not report exactly the number of study participants who were vaccine hesitant, one out of ten of the study participants disagreed that COVID-19 vaccine was safe, while 21 percent disagreed that COVID-19 vaccine is effective, yet a whopping 40 percent of the study participants had doubts of COVID-19 vaccine safety despite the WHO recommendation of it use.

Eight out of ten study participants who took the vaccine reported side effects, of which more than half claimed they had moderate to severe side-effects. Our findings point to severe headaches as the one side effect reported by more than half of the respondents who took the COVID-19 vaccine. Although some considered their experiences as mild, the presence of side effects appears to mitigate the odds of achieving an optimal level of inoculation against the novel coronavirus, and this is consistent with earlier studies that suggest that Ghanaian healthcare professionals showed mild vaccine side effects that resolved with or without interventions [25]. Structured education should follow vaccination in the future, giving the chances of side effects.

The subject of mistrust of vaccine versus expert recommendation and its implication cannot be overemphasized. Healthcare workers had no full trust in the COVID-19 vaccine safety in spite of experts' recommendations and this probably explains the declining figures at the level of the second dosing. In our analysis, a little over half of the study participants took a second dose. For a full protection, most COVID-19 vaccine require at least two doses jab to be effective, and what this suggest is that, unless efforts are made to dose population with a single dose, herd immunity may be difficult to attain despite higher figures of first doses of vaccination.

Our study also observed self-mistrust and uncertainty surrounding vaccine safety and effectiveness. This finding resonates with previous vaccine safety and effectiveness studies

as the one reason for vaccine hesitancy [9,17,26]. We found that perception of COVID-19 vaccine safety, perceived disease seriousness, lack of trust in expert views and previous vaccination experience influenced the odds of taking the COVID-19 vaccine, and these were statistically significant, p = 0.001, p = 0.008, p = 0.015, p = 0.001, respectively. The findings are similar to an earlier study in Iraq [27] and in the U.S. [28,29] that found that perceived risk of infection was a significant predictor for vaccination acceptance. Nevertheless, the findings contrast with earlier views, canvassing expert recommendations as a catalyst to whip up vaccine uptake [25,30]. Known comorbidity and professional category negatively influenced the uptake of COVID-19 vaccines, which were statistically significant.

## Limitation and strength

The study design could not include data from a more expansive setting of Ghanaian healthcare workers, which may have skewed the results, reflecting only the views of health workers in the middle belt of Ghana. Also, study participants religion was nine out of ten Christians, and this perhaps skewed the perceptions and beliefs expressed in the findings thereof. However, the use of probability sampling somewhat offsets the weaknesses of the study design. Using one-time data with no validated survey data of national status means the findings could not account for subsequent knowledge that may have been influenced by on-going public education, including healthcare workers. Nevertheless, the findings of this study add to the literature on COVID-19 vaccine perception and uptake from a deprived setting.

## Conclusions

Perceived non-severity of COVID-19 infection, vaccine safety concerns, severity of vaccine side effects, and the lack of trust in expert recommendations affects the uptake of the COVID-19 vaccine among healthcare workers. It is recommended that the Ministry of Health and Ghana Health Service provide tailor-made education on COVID-19 vaccination to deal with lingering myths and misconceptions even among healthcare workers to improve COVID-19 vaccine uptake as a sure means of achieving herd immunity against the resurgence of the coronavirus disease.

## Supporting information

**S1 Table. Chi-square test of associations between sociodemographic characteristics and health worker vaccine uptake**
(DOCX)

## Author contributions

**Conceptualization:** Emmanuel K. Gelyi.

**Data curation:** Emmanuel K. Gelyi, Mary Rachael Kpordoxah.

**Formal analysis:** Emmanuel K. Gelyi, John Azaare, Nana Kobea Bonso, Mary Rachael Kpordoxah, Gifty Apiung Aninanya.

**Methodology:** John Azaare, Nana Kobea Bonso, Gifty Apiung Aninanya.

**Supervision:** John Azaare, Gifty Apiung Aninanya.

**Writing – original draft:** Emmanuel K. Gelyi.

**Writing – review & editing:** John Azaare, Nana Kobea Bonso, Mary Rachael Kpordoxah, Gifty Apiung Aninanya.

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
