## [Decision Letter · Decision Letter 0]

11 Nov 2024

PONE-D-24-31145Impact of perceived factors of coronavirus infection on COVID-19 vaccine uptake among health care workers in Ghana - evidence from a cross-sectional analysis.PLOS ONE

Dear Dr. Azaare,

Thank you for submitting your manuscript to PLOS ONE. After careful consideration, we feel that it has merit but does not fully meet PLOS ONE’s publication criteria as it currently stands. Therefore, we invite you to submit a revised version of the manuscript that addresses the points raised during the review process.

We look forward to receiving your revised manuscript.

Kind regards,

Seth Agyei Domfeh, PhD

Academic Editor

PLOS ONE

Journal Requirements:

1. Please ensure that your manuscript meets PLOS ONE's style requirements, including those for file naming. The PLOS ONE style templates can be found at https://journals.plos.org/plosone/s/file?id=wjVg/PLOSOne_formatting_sample_main_body.pdf and https://journals.plos.org/plosone/s/file?id=ba62/PLOSOne_formatting_sample_title_authors_affiliations.pdf.

Reviewers' comments:

Reviewer's Responses to Questions

**Comments to the Author**

1. Is the manuscript technically sound, and do the data support the conclusions?

Reviewer #1: Yes

Reviewer #2: Yes

2. Has the statistical analysis been performed appropriately and rigorously? 

Reviewer #1: Yes

Reviewer #2: Yes

3. Have the authors made all data underlying the findings in their manuscript fully available?

Reviewer #1: Yes

Reviewer #2: Yes

4. Is the manuscript presented in an intelligible fashion and written in standard English?

Reviewer #1: Yes

Reviewer #2: Yes

5. Review Comments to the Author

Reviewer #1: Well done in describing Impact of perceived factors of coronavirus infection on COVID-19 vaccine uptake among health care workers in Ghana - evidence from a cross-sectional analysis, the authors should be congratulated

Reviewer #2: I have added my comments to a copy of the manuscript attached to this. This was easier for my review as the manuscript was not line numbered to facilitate easy review. Please find attached for your attention.

6. PLOS authors have the option to publish the peer review history of their article (what does this mean? ). If published, this will include your full peer review and any attached files.

**Do you want your identity to be public for this peer review?** For information about this choice, including consent withdrawal, please see our Privacy Policy .

Reviewer #1: No

Reviewer #2: **Yes: ** George Gyamfi-Brobbey

---

## [Author Response · Author response to Decision Letter 1]

30 Dec 2024

REBUTTAL LETTER

PLOSE ONE JOURNAL

Dear Editor,

RESPONSE TO REVIEWER COMMENTS

We thank you and the reviewers for taking time to read and provide comments to our manuscript. The comments were indeed helpful and will add clarity to the paper and enrich the discourse around COVID-19 vaccine uptake. Reviewer 1 had no comments requiring response and we are most grateful for the congratulatory message. Reviewer made valuable comments and we have attempted to respond to all the comments and have now attached our response to each comment as captured below. We hope that you will find them satisfactory and that our manuscript will receive a favorable reviewer outcome.

Regards.

Signed

John Azaare (PhD)

(corresponding author)

Reviewer 2 Comments:

Point by Point Response to reviewer comments.

1. Comment: “Any of the COVID-19 Vaccines administered, I guess?”

Response: Yes, any of the administered Covid 19 vaccines.

2. Comment: “It would be better to state the names of the vaccines rather than the manufacturers.”

Response: Well, manufacturers produced specific covid-19 vaccines at the time, and therefore names of manufactures represented the vaccine type. This study took no record of the specific generic name of the vaccines.

3. Comment: “This is currently not the case as COVID-19 is no longer a pandemic. Amend the statement.”

Response: We agree with the reviewer. Sentence has been amended in the introduction section of the manuscript to reflect current situation.

4. Comment: “Please rewrite this statement in two sentences to make it clearer.”

Response: Sentence has been revised and written into two. We hope this makes reading much clearer.

5. Comment: “Compared to the case in Ghana, it would be good to state the exact jurisdictions being referred to.”

Response: This sentence has been revised, and the exact places stated. We thank the reviewer for the observation.

6. Comment: “There were purported conspiracy theorists and anti-vaxxers whose activities influenced vaccine update. How did they change/affect the drive to increase vaccine uptake. Any reference to support this?”

Response: The sentence referred to by the reviewer is a general observation that we made taking into accounts the uncertainty that arose at the advent of the COVID-19 pandemic. We do not find the statement so much authoritative or a statement of fact to require a reference. However, the sentence has been revised, perhaps, to allow for much simpler reflection of our intentions, and we hope this will address the concerns of the reviewer.

7. Comment: “I am not sure this statement is clear.”

Response: Statement has been reviewed under ‘ethical approval and consent to participate section. We hope this addresses the reviewer’s concern.

8. Comment: “Did the study consider getting views and responses from non-Christians? I know the respondents were randomized and the Mampong area is largely made up of Christians, it would be good to hear the opinions of the people belonging to different faiths or practices.”

Response: As presented in Table 1 of the manuscript, 95.8% of respondent were Christians, consistent with the reviewer assertion about the dominant religion of the study area. Since this study sampled randomly, there was no to segregate the opinion of other religion participants different from Christians. We have thus included into the limitation section as a caution to guide interpretation and application of the results given the skewed nature of the Christians beliefs of the study participants.

9. Comment: “Were participants given the opportunity to choose which vaccine to take?”

Response: We than the reviewer for asking this question. While the design of study failed to ask participants whether there was an option to choose which brand they wanted, it is a given that health workers necessarily did not take the vaccine from one site i.e. the study with multiple brands, and therefore, choice of vaccine may have been limited and influenced by availability. This observation is reflected in the discussion section of the manuscript.

10. Comment: “Was there any reported hesitancy in vaccine uptake among the participants, if there were, how many?”

Response: While this study did not report exactly the number of study participants who were vaccine hesitant, the study did report (Table XX) that 10% of participant disagree that the vaccine was safe, while 21% disagree that COVID-19 was safe, with a whopping 40% of the study participants had doubts of safety despite WHO recommendations for its use. This statement is reflected in the discussion section.

11. Comment: “Please is there any particular reason to this?”

Response: Reviewer query is in reference to an earlier study finding that we referred to in our discussion section. It is difficult for us to the query by attributing reasons to those finding since it is not part of the findings of the current study.

12. Comment: “Can you please look at this statement again. The meaning is not clear.”

Response: statement has been revised and re-written in the discussion section appropriately. We hope it addresses the concerns of the reviewer.

13. Comment: “Can you please provide a web address for Reference No. 2. Also, the style of referencing is not uniform throughout the list.”

Response: We thank the reviewer for his observation on the referencing format. This have been revised for reference no. 15. However, the web address for reference no. 2 is; https://www.who.int/emergencies/diseases/novel-coronavirus-2019 and this has been captured in the reference section. It is not clear to us whether the rewiewer is looking for something different from this. If there are more specifics than what has been addressed, the reviewer may specify and we are glad to address them.

Once again, we are most graeful to the reviwers for the time taken to read and provide valauble comments to the manuscript.

Thank you.

---

## [Decision Letter · Decision Letter 1]

21 Jan 2025

Impact of perceived factors of coronavirus infection on COVID-19 vaccine uptake among health care workers in Ghana - evidence from a cross-sectional analysis.

PONE-D-24-31145R1

Dear Dr. Azaare,

We’re pleased to inform you that your manuscript has been judged scientifically suitable for publication and will be formally accepted for publication once it meets all outstanding technical requirements.

Kind regards,

Seth Agyei Domfeh, PhD

Academic Editor

PLOS ONE

---

## [Editor Report · Acceptance letter]

PONE-D-24-31145R1

PLOS ONE

Dear Dr. Azaare,

I'm pleased to inform you that your manuscript has been deemed suitable for publication in PLOS ONE. Congratulations! Your manuscript is now being handed over to our production team.

Kind regards,

on behalf of

Dr. Seth Agyei Domfeh

Academic Editor

PLOS ONE